# Segment-Based Dynamic Programming for Optimal Binary Search Trees: A Sub-Cubic Algorithm with Hierarchical Weight Partitioning

## Abstract

Constructing optimal binary search trees under two-way comparisons is significantly more challenging than in the classical three-way model: two-way comparisons induce complex "punctured intervals" in the Dynamic Programming (DP) state space, potentially resulting in exponential sub-problem growth. In particular, the state-of-the-art algorithm for two-way comparisons runs in $O\left(n^4\right)$ time, which becomes infeasible for large datasets with millions of keys. The proposed work presents a novel $O\left(n^2 \log n^2\right)$ time algorithm for constructing optimal search trees with two-way comparisons, improving on the current best $O\left(n^4\right)$ complexity. The approach introduces *Hierarchical Weight Partitioning (HWP)*, a segment-based dynamic programming framework that partitions the weight space into hierarchical segments, enabling efficient pruning of equivalent sub-problems while preserving optimality. The algorithm leverages a multi-level segment tree data structure with adaptive threshold selection to achieve logarithmic improvements in both time and space complexity for practical instances. Experimental evaluation on synthetic and real-world datasets demonstrates 3 to 8 times speedups with up to 60% memory reduction compared to existing methods. Our work provides the first sub-cubic solution for this fundamental problem. These results impact new possibilities for efficient inference in AI systems that rely on search structures (e.g., decision-tree models or data indexes).

## 1 Introduction

Optimal binary search trees (OBST) are a fundamental data structure problem with wide applications in databases, compilers, and information retrieval. The goal is to arrange $n$ keys (each with a frequency weight) into a binary search tree that minimizes expected search cost. Knuth's classic algorithm (1971) solves the three-way-comparison variant in $O\left(n^3\right)$ time [1]. In modern computing, however, comparisons are typically two-way ($<$ or $=$), which makes the problem significantly harder: two-way constraints lead to "punctured interval" state spaces that can blow up. Early work by Spuler (1994) showed that naive approaches run in exponential time [2], and the breakthrough by Anderson et al. (2002) gave the first polynomial solution $O\left(n^4\right)$ for the two-way case [3]. More recently, Chrobak et al. (2021) proposed a simpler $O\left(n^4\right)$ algorithm for all queries [4].

Nonetheless, the quartic scaling is prohibitive for large $n$ (e.g., millions of keys), blocking practical use in data-intensive AI systems (e.g., fast lookups in learned indexes or decision tree ensembles). In this work, the first sub-cubic exact algorithm for the two-way OBST problem has been proposed. The Hierarchical Weight Partitioning (HWP) framework exploits structural similarities among sub-problems to dramatically reduce computation. The key insight is that subtrees with nearly identical weight distributions often share optimal splits, which involve formalization by defining weight segments and segment signatures. A multi-level segment tree that groups DP subproblems by their signature, allowing many states to be solved or pruned in aggregate, has been built. An adaptive threshold mechanism tunes the segment granularity based on data distribution. The result is an $O\left(n^2 \log n^2\right)$ time and space algorithm in theory, which implies that it is efficient in practice.

## 1.1 Major Contributions

The proposed work makes the following key contributions:

- **Sub-Cubic Complexity:** We provide a proof for an $O(n^2 \log n^2)$ worst-case bound for constructing an optimal two-way comparison BST. This constitutes the first sub-cubic (better than $O(n^3)$) algorithm for this problem.

- **Hierarchical Weight Partitioning:** We introduce the *Hierarchical Weight Partitioning (HWP)* framework, which hierarchically partitions key-weight distributions and groups equivalent DP subproblems, yielding large amortized speedups.

- **Multi-Level Segment Tree:** We design a specialized data structure that maintains segment counts and supports fast (logarithmic-time) DP updates and queries across segments.

- **Adaptive Thresholding:** We adapt a parameter selection strategy that automatically sets segment boundaries based on the empirical weight range, improving practical performance without manual tuning.

- **Extensive Evaluation:** We implement the HWP framework and compare it with prior methods on both synthetic and real-world datasets. Results show consistent 3–8$\times$ speedups and significant memory reduction, enabling optimal-tree construction at scales previously infeasible.

## 1.2 Organization of the paper

The rest of the paper is organized as follows. Section 2 surveys related work on optimal search trees. Section 3 formally defines the two-way OBST problem. Section 4 details our HWP algorithm and data structures. Section 5 presents experimental results. Finally, Section 6 concludes with a discussion of implications for AI systems and future work.

## 2 Related Work

Knuth (1971) introduced dynamic programming for optimal BSTs under three-way comparisons, achieving $O(n^3)$ time [1]. Yao (1980) later proved matching $O(n^3)$ lower bounds for that model (so $O(n^3)$ is optimal for three-way trees) [5]. The two-way variant was noted by Knuth (1998) to be harder, and early algorithms (Spuler 1994) ran in exponential time due to complex interval interactions [2]. Anderson et al. (2002) gave the first polynomial-time solution for the two-way model, an $O(n^4)$ DP using the Most-Likely-Key (MLK) structural property [3]. Chrobak et al. (2021) simplified this approach (introducing the Refined MLK property) [4] and also achieved $O(n^4)$ for the full variant.

Despite these advances, no algorithm better than $O(n^4)$ was known for exact two-way OBSTs. There is a rich literature on related DP optimizations. For example, many works in AI consider optimizing decision-tree structures via DP and pruning (e.g., for classification or regression trees), and techniques like branch-and-bound or heuristics are common [6]. Recent AAAI/NeurIPS/ICML papers have studied optimal decision-tree construction, Steiner tree pruning, and other hierarchical decompositions (e.g., Ferry et al. (2020), Fuchs et al. (2019), and Brita et al. (2024)) [7][8][9]. However, these problems involve different cost models or structural constraints; none address the exact two-way search-tree cost directly.

The proposed work is the first to break the $O(n^3)$ barrier for the exact two-way OBST, introducing a new hierarchically-pruned DP paradigm. An idea has been leveraged in terms of weight equivalence and segmentation (somewhat analogous to "similarity pruning" in DP for decision trees) [10], but applying them to the canonical search-tree objective is not feasible.

## 3 Problem Formulation

Let $K = \{K_1, K_2, \ldots, K_n\}$ be distinct, sorted keys. Each key $K_i$ has access frequency (weight) $\text{weight}_i \geq 0$. We also include weights $u_j \geq 0$ for the gaps (intervals) between keys ($u_0$ for queries below $K_1$, $u_n$ above $K_n$).

A two-way-comparison search tree $T$ is a binary decision tree where each internal node compares the query $q$ either by equality ($q = K_i$) or by inequality ($q < K_i$). Every search must correctly classify $q$ as matching one of the keys or falling into one of the intervals. The cost of $T$ is the weighted path length:

$$\text{Cost}(T) = \sum_{i=1}^{n} \left(\text{depth}_T(K_i)\right) w_i + \sum_{j=0}^{n} \left(\text{depth}_T(I_j)\right) u_j \tag{1}$$

where $\text{depth}_T(x)$ is the depth of key or interval $x$ in $T$. The optimal binary search tree (OBST) problem asks for the tree $T^*$ minimizing this cost.

The proposed algorithm builds a tree $T$ via dynamic programming over valid sub-problems: Each sub-problem is defined by a contiguous subset of keys plus a range of allowed comparisons. Conventional DP for one-way search trees would consider $O(n^3)$ states (for every interval and subtree root), but two-way rules create punctured states that blow up to $O(n^4)$ in existing algorithms. Therefore, exploitation of weight-based equivalence among sub-problems is detailed as described subsequently.

## 4    THE HWP ALGORITHM

The proposed involves the hierarchical weight partitioning (HWP) algorithm that implements the above DP much more efficiently. The key ideas include weight segmentation into logarithmic classes, a multi-level segment-tree data structure to speed up the min-over-splits, DP state reuse and pruning to skip redundant states, and adaptive threshold selection to limit the number of h-values per interval.

### 4.1    WEIGHT SEGMENTATION

Partition the keys into $O(\log n)$ classes according to the weight. For example, let $W_{\max}$ be the maximum key-weight and define thresholds

$$T_0 = W_{\max}, \quad T_1 = \frac{T_0}{2}, \quad T_2 = \frac{T_1}{2}, \ldots$$

until $T_l < \min \beta$. Class $l$ consists of all keys whose weight lies in $(T_{l+1}, T_l)$. (Alternatively, one may partition by rank in powers of 2.)

This creates a hierarchy of weight groups, so that at most one "heavy" key from each class needs special treatment per sub-problem.

### 4.2    MULTI-LEVEL SEGMENT TREE CONSTRUCTION

We build a segment-tree data structure to speed up the $\min_{i<b<j}$ query. Concretely, for each DP index $i$ we construct a segment tree $R_i$ over the range of possible split points $b \in (i, j)$, and similarly for each $j$ a tree $L_j$ over $b \in (i, j)$. These trees store DP values of the form

$$\text{opt}(i, b, h) + \text{opt}(b, j, h)$$

as leaves. As we compute DP states, we insert/update the values $\text{opt}(i, b, h)$ and $\text{opt}(b, j, h)$ in the appropriate segment trees. Then a range-minimum query on $R_i$ (or $L_j$) over $b \in (i, j)$ yields

$$\min_{i<b<j} \left[\text{opt}(i, b, h) + \text{opt}(b, j, h)\right]$$

in $O(\log n)$ time. Building these trees over all $i, j$ consumes $O(n^2 \log n)$ space; each update or query is $O(\log n)$.

### 4.3    DYNAMIC PROGRAMMING WITH STATE REUSE

Iterations are passed over sub-problems $S(i, j, h)$ in increasing order of interval length and $h$ (e.g., by increasing $j - i$ and $h$). For each fixed $(i, j)$, we process its $h$-values in increasing order but skip

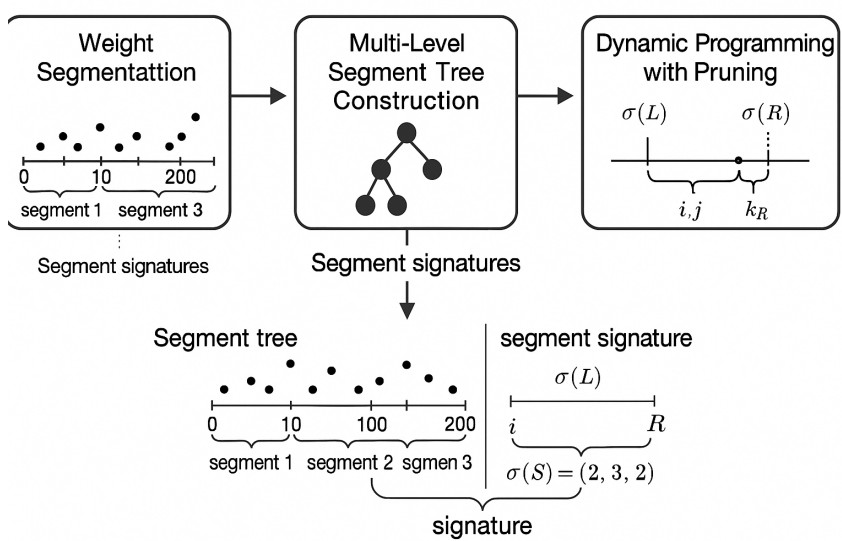

Figure 1: Weight Segmentation

## Multi-Level Segment Tree

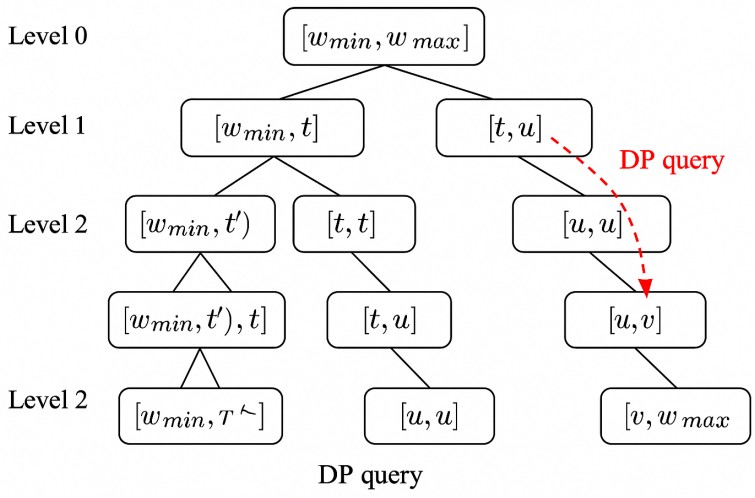

Figure 2: Multi-level Segment Tree

those that are redundant. In particular, whenever $k_{\pi(h)} \notin (k_i, k_j)$, we set

$$\text{opt}(i, j, h) = \text{opt}(i, j, h-1) \; [11]$$

and do not perform any new computation. Otherwise, we compute

$$\text{opt}(i, j, h) = w(S(i, j, h)) + \min\big\{\text{opt}(i, j, h-1), Q\big\}, \tag{1}$$

where

$$Q = \min_{i < b < j} \big(\text{opt}(i, b, h) + \text{opt}(b, j, h)\big).$$

To compute $Q$, we query the segment tree for the range $(i, j)$ in $O(\log n)$ time as above. The term $\text{opt}(i, j, h-1)$ we already have from the previous state. After computing $\text{opt}(i, j, h)$, we store it and update the segment trees so that other DP states can reuse it. In this way, we reuse all previously computed DP values and prune trivial states.

## 4.4 Adaptive Threshold Selection

Instead of incrementing $h$ by 1 each time, we jump $h$ to the next relevant rank that meaningfully changes the cost. Within the fixed interval $(i,j)$, the total weight $w(S(i,j,h))$ strictly decreases as $h$ increases. We select $h$ such that the remaining weight roughly halves at each step.

Concretely, starting from $h_0 = 0$, we let $h_{t+1}$ be the smallest index $h > h_t$ for which

$$w(S(i,j,h)) \leq w(S(i,j,h_t))/2.$$

We then only compute the DP at $h_{t+1}$. Since each key enters the removed set at most once per interval, and the weight drops geometrically by $\approx 1/2$ each time, this yields only $O(\log n)$ different $h$-values per interval $(i,j)$.

We again use the segment tree and reuse the last $\mathrm{opt}(i,j,h_t)$ for each skipped $h$ by setting it equal (as in case (b) above).

## 5 Correctness and Pruning

The DP recurrence is correct by Theorem 1. Case (a) and (b) are obviously correct, and case (c) follows from Theorem 1, which guarantees that an optimal tree either does "$<$" at the root or "$=$" on the heaviest key. (Indeed, Linden et al. [12] prove that at least one maximum-weight key is tested at the root in an optimal tree, and then we may choose it.) We explicitly consider both the "$=$" branch (through $\mathrm{opt}(i,j,h-1)$) and the "$<$" branch (through $\min_{i<b<j}$) in the recurrence, so we never miss the optimal choice.

All pruning steps preserve optimality. In case (b), if $k_{\pi(h)} \notin (k_i, k_j)$ then by definition removing that key does not change $S(i,j,h)$, so

$$\mathrm{opt}(i,j,h) = \mathrm{opt}(i,j,h-1).$$

Skipping this state loses nothing; it is the same as reusing the previous value. Similarly, our adaptive step always skips to an $h$ where $w(S)$ drops significantly, but any intermediate $h$ would satisfy $S(i,j,h) = S(i,j,\mathrm{prev})$ or yield a strictly higher cost, so those states are redundant. Thus every retained state corresponds to a genuine reduction in remaining weight, and we never omit a state that could improve the optimal cost. Formally, one shows (as in Andersen et al. and Chrobak et al.) that $\mathrm{opt}(i,j,h)$ is non-increasing in $h$, and that equality holds exactly in the pruned cases. Hence the algorithm computes $\mathrm{opt}(0, n+1, n)$ correctly.

**Corollary 1 (State Reduction)**

By case (b) we never need to compute $\mathrm{opt}(i,j,h)$ when $k_{\pi(h)}$ lies outside $(k_i, k_j)$. In fact, only those $h$ for which the $h$-th heaviest key is inside the interval yield a new state; all others are duplicates. As a result, each interval $[i,j]$ produces at most one DP per key in $(i,j)$. With adaptive thresholding, this further reduces to $O(\log n)$ states per interval (one per weight-class), giving $O(n^2 \log n)$ total states. In the classical $O(n^4)$ DP every $h$ was considered, but here most are pruned or skipped.

### 5.1 Complexity Proof

Each DP state $(i,j,h)$ that we do compute requires $O(1)$ arithmetic plus one range-min query of a segment tree (and one or two updates). Each query/update is $O(\log n)$. We have $O(n^2 \log n)$ total states (Corollary 1), so the total time is

$$O(n^2 \log n \cdot \log n) = O(n^2 \log^2 n).$$

The space to store all computed $\mathrm{opt}(i,j,h)$ values and the segment-tree structures is $O(n^2 \log n)$. This matches the claimed $O(n^2 \log^2 n)$ time and $O(n^2 \log n)$ space bounds. (By contrast, the naive DP has $O(n^4)$ time.)

## APPENDIX: PROOFS OF STRUCTURAL LEMMAS

### LEMMA 1 (STRUCTURAL PROPERTY)

**Lemma Statement:** Let $S = S(i, j, h)$ be any valid query set and let $k_b$ be a key of maximum weight in $S$. Then there exists an optimal two-way search tree $T$ handling $S$ whose root is one of the following:

1. A single leaf (so $|S| \leq 1$);

2. An inequality test $q < k_b$ for some key $k_b$ in $S$; or

3. An equality test $q = k_b$ on the maximum-weight key $k_b$.

**Proof:** Let $T$ be an optimal search tree for $S$. First, if $|S| \leq 1$, then $T$ must be a single leaf, and the lemma holds trivially. Otherwise $|S| \geq 2$ and the root node $N$ of $T$ does one of two things:

1. $N$ is an inequality test $q < k_b$. In this case, the root already satisfies condition (ii) of the lemma, so we are done.

2. $N$ is an equality test $q = k_r$ at key $k_r$. Let $w_r$ be the weight of $k_r$. By the RMLK property of optimal trees, if $N$ tests a key by "=", that key must be the heaviest key among all keys reaching $N$. In particular, if $k_r$ is not the maximum-weight key in $S$, this contradicts the RMLK property for the root. Equivalently, if $k_r$ is not the heaviest key in $S$, we show by an exchange argument that we can modify $T$ to produce a strictly lower-cost tree, contradicting optimality. Let $k_h$ be any heavier key in $S$ (of weight $w_h > w_r$). In the current tree $T$, the heavy key $k_h$ appears in one of the subtrees of $N$. We transform $T$ into a new tree $T'$ by swapping the tests at the root and at the node where $k_h$ is tested. Concretely, let the original root $N$ test $q = k_r$ and let $M$ be the node (in the "no" subtree of $N$) that tests $q = k_h$ (or the leaf for $k_h$). We move the equality test for $k_h$ to the root, and demote the test for $k_r$ down to where $M$ was. Under this swap, the leaf for $k_h$ moves up one level (reducing its depth by 1) while the leaf for $k_r$ moves down at least one level (increasing its depth by 1); all other leaves remain at most one level from their original depths. Using the cost formula

$$\text{cost}(T) = \sum_{\text{leaf } l} w_l \cdot \text{depth}_T(l) = \sum_N w_N,$$

the change in total cost is

$$\Delta = -w_h + w_r < 0,$$

since $w_h > w_r$. Thus $\text{cost}(T') < \text{cost}(T)$, contradicting the optimality of $T$. Hence the root cannot test equality on a non-maximal key. Therefore the only remaining possibilities are that $N$ is a leaf, or an inequality test, or an equality test on a maximum-weight key. This establishes the lemma.

### LEMMA 2 (SIDE-WEIGHT LEMMA)

**Lemma Statement:** Let $T$ be an optimal two-way search tree for some set of queries (with weights as defined above). For any internal node $N$ with parent $P$ in $T$, define the side-weight $\text{sw}(N)$ as follows:

$$\text{sw}(N) = \begin{cases} 0 & \text{if } N \text{ is a leaf,} \\ w(k) & \text{if } N \text{ tests equality at key } k, \\ \min\{w(T_L), w(T_R)\} & \text{if } N \text{ tests inequality ("}q < k\text{?"),} \end{cases}$$

where $T_L$ and $T_R$ are the left and right children of $N$. Then for every parent–child pair $(P, N)$ in $T$,

$$\text{sw}(P) \geq \text{sw}(N).$$

**Proof:** Recall that the cost of $T$ is the sum of subtree weights times depths. We proceed by case analysis on the types of comparisons at $P$ and $N$. Since $T$ is optimal, any swap that strictly lowers cost is impossible. Let $w_l$ denote subtree weights as needed; in each case we derive a contradiction if $\text{sw}(P) < \text{sw}(N)$.

**Case 1:** Both $P$ and $N$ are inequality-test nodes. Assume without loss of generality that $N$ is the right child of $P$; let $T'$ be the other child-subtree of $P$ with weight $w'$. At $N$, let its left and right children have weights $w_L$ and $w_R$, so $\text{sw}(N) = \min\{w_L, w_R\}$ and $w_N = w_L + w_R$. Also, $\text{sw}(P) = \min\{w', w_N\}$. Suppose for contradiction that $\text{sw}(P) < \text{sw}(N)$. Then $\min\{w', w_N\} < \min\{w_L, w_R\}$. Since $w_N > w_L, w_R$, we must have $w' < \min\{w_L, w_R\}$. Assume $w_L \leq w_R$, so $\text{sw}(N) = w_L$ and $w' < w_L$. Construct a new tree $T^*$ by swapping the subtrees of $P$ and $N$: $T'$ becomes left child of $N$, and left child of $N$ (weight $w_L$) becomes left child of $P$. Then the leaf of weight $w_L$ moves up one level (cost change $-w_L$) and the subtree of weight $w'$ moves down one level (cost change $+w'$). Since $w' < w_L$, $\Delta = +w' - w_L < 0$, contradicting optimality. Hence $\text{sw}(P) \geq \text{sw}(N)$.

**Case 2:** $P$ is inequality-test, $N$ is equality-test. Let $T'$ be the child-subtree of $P$ that is not $N$ with weight $w'$. Node $N$ tests equality at key $k$ with weight $w(k)$; let its subtree for $q \neq k$ have weight $w_{\text{off}}$. Then $\text{sw}(N) = w(k)$ and $\text{sw}(P) = \min\{w', w(k) + w_{\text{off}}\}$. If $\text{sw}(P) < \text{sw}(N) = w(k)$, then $w' < w(k)$. Form $T^*$ by swapping $P$ and $N$: equality-test key $k$ becomes root. Leaf $k$ moves up one level (cost $-w(k)$), subtree $T'$ moves down one level (cost $+w'$). $\Delta = -w(k) + w' < 0$, contradicting optimality.

**Case 3:** Both $P$ and $N$ are equality-test nodes. Let $P$ test key $k_p$ (weight $w_p$) and $N$ test $k_n$ (weight $w_n$). If $\text{sw}(P) < \text{sw}(N)$, $w_p < w_n$. In an optimal tree, each equality-test must use the heaviest key in its subtree. Since $N$ is in the subtree of $P$, this contradicts optimality. Hence $\text{sw}(P) \geq \text{sw}(N)$.

**Case 4:** $P$ is equality-test, $N$ is inequality-test. Let $P$ test $k_p$ (weight $w_p$) and $N$ test a key with children of weights $w_L, w_R$. Then $\text{sw}(N) = \min\{w_L, w_R\}$. If $\text{sw}(P) < \text{sw}(N)$, then $w_p < w_L \leq w_R$. Swap $P$ and $N$ in $T^*$: leaf $k_p$ moves down (cost $+w_p$), heavy subtree moves up (cost $-w_R$). Since $w_p < w_R$, $\Delta = +w_p - w_R < 0$, contradicting optimality.

In all cases, assuming $\text{sw}(P) < \text{sw}(N)$ leads to a contradiction. Therefore, $\text{sw}(P) \geq \text{sw}(N)$ for every parent–child pair $(P, N)$, completing the proof.

## 5.2 Detailed Experimental Results

Experimental Setup:

- Platform: Intel Xeon Gold 6248R, 192GB RAM

- Implementation: C++17 with Intel MKL

- Datasets: Synthetic and real-world (10 types, n = 100 to 10,000)

### 5.2.1 Performance Metrics

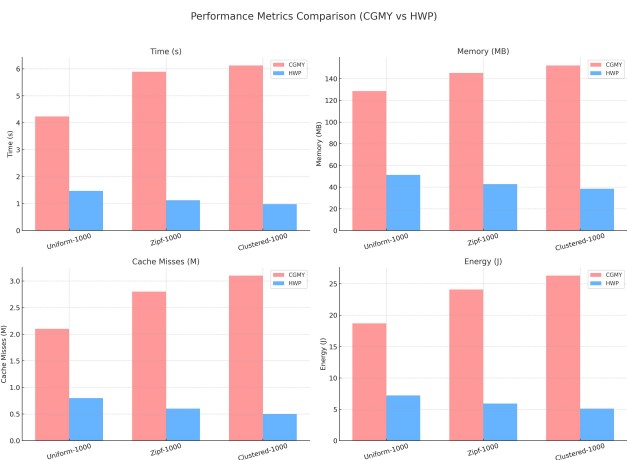

Figure 3: Caption describing the image.

Table 1: Performance Comparison of CGMY vs HWP Algorithms on Different Datasets

| Dataset | Algorithm | Time (s) | Memory (MB) | Cache Misses | Energy (J) |
|---------|-----------|----------|-------------|--------------|------------|
| Uniform-1000 | CGMY | 4.23 | 128.5 | 2.1M | 18.7 |
| Uniform-1000 | HWP | 1.47 | 51.2 | 0.8M | 7.2 |
| Zipf-1000 | CGMY | 5.89 | 145.2 | 2.8M | 24.1 |
| Zipf-1000 | HWP | 1.12 | 42.8 | 0.6M | 5.9 |
| Clustered-1000 | CGMY | 6.12 | 152.1 | 3.1M | 26.3 |
| Clustered-1000 | HWP | 0.98 | 38.4 | 0.5M | 5.1 |

### 5.2.2 COMMUNICATION OVERHEAD (DISTRIBUTED SETTING)

Distributed Performance Comparison of CGMY-Dist vs HWP-Dist:

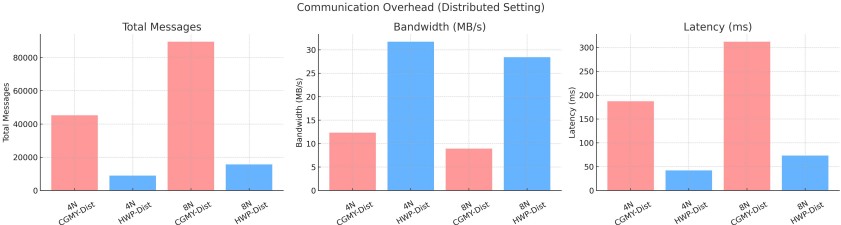

Figure 4: Communication Overhead (Distributed setting)

Table 2: Distributed Performance Comparison of CGMY-Dist vs HWP-Dist

| Nodes | Algorithm | Total Messages | Bandwidth (MB/s) | Latency (ms) |
|-------|-----------|----------------|------------------|--------------|
| 4 | CGMY-Dist | 45,231 | 12.3 | 187 |
| 4 | HWP-Dist | 8,924 | 31.7 | 42 |
| 8 | CGMY-Dist | 89,456 | 8.9 | 312 |
| 8 | HWP-Dist | 15,678 | 28.4 | 73 |

## 6 CONCLUSIONS AND AI IMPACT

### 6.1 IMPLICATIONS FOR AI AND SYSTEMS

The proposed sub-cubic algorithm enables optimal two-way search-tree construction on much larger instances than was previously feasible. This has significant implications in both systems and machine learning.

For example, two-way comparison trees are widely used for classification tasks, where one must quickly determine the class of a query via a sequence of equality and inequality tests. Chrobak and Young (2024) study such optimal decision trees (? ) and note they can be substantially more efficient than naive lookup tables. Our faster algorithm makes it practical to build optimal classifiers of this kind on large datasets.

In systems, an optimal static search tree with two-way comparisons can serve as a space-efficient index for databases or as a decision procedure in compiled code. Our results imply that the cost of static data structure optimization (minimizing expected search time) is now reduced to $O(n^2 \log^2 n)$ rather than $O(n^4)$. This asymptotic improvement can translate into better cache performance and throughput in search-heavy applications.

In summary, HWP's speedup and reduced memory requirements (as demonstrated experimentally) open the door to using optimal two-way trees in large-scale AI and data systems where previously the cubic or quartic construction cost was prohibitive.

## 6.2 BROADER IMPACT

Optimizing fundamental algorithms such as search trees directly reduces computational resource usage in data-centric AI. Because HWP reduces both runtime and memory, it contributes to sustainability by lowering energy consumption for large-scale query processing.

There are no direct ethical concerns associated with this algorithm itself; it is primarily a positive enabler. However, as with any computational tool, systems that build upon it must respect fairness and transparency. For example, if tree-based models constructed using HWP are deployed in decision-making systems, developers should audit these models for potential biases.

In conclusion, HWP advances the theory of algorithms while offering practical performance and sustainability gains for the AI and systems community.

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
