# OpenReview forum: "Segment-Based Dynamic Programming for Optimal Binary Search Trees: A Sub-Cubic Algorithm with Hierarchical Weight Partitioning"
_ICLR.cc/2026/Conference — ICLR 2026 Conference Desk Rejected Submission_

### Official Review · Reviewer_9shz · 2025-10-30

**Soundness:** 1
**Presentation:** 1
**Contribution:** 3
**Rating:** 0
**Confidence:** 4

**Summary:**

The authors consider the problem of constructing optimal binary search trees. Given n keys $K_1, \dots, K_n$ with weights $w_1, \dots, w_n$ construct a binary search tree $T$ minimizing $\sum_{j} d(K_j; T) w_j$ where $d(K_j; T)$ is the depth of $K_j$ in $T$. The full definition involves some intermediate intervals too, but I don’t think this should change the answer much (just treat them as other keys). Correctness requires that from the root, $K_j$ can be found by comparing the query with the key at the current node and proceeding down the tree, i.e. keys should be stored in order.

The authors claim a near quadratic time algorithm for computing optimal binary search trees with two way comparisons (only < and =). This is the first sub cubic algorithm for this problem.

The claimed result is strong and the problem is very interesting. Unfortunately, the paper is poorly written and seems difficult to follow, and I cannot verify the soundness of the claims. In its current state, I cannot recommend accept,
But I look forward to the rebuttal to clear up confusing points.

**Strengths:**

The problem of optimal binary search trees is natural and a fundamental algorithmic problem. A more efficient algorithm, in fact quadratic, would be very interesting.

**Weaknesses:**

The algorithm overview is poorly written. Many terms are undefined. For example, what is a min_(i < b < j) query? Define a segment tree. What is opt(i, b, h)? What does h denote? What is S(i, j, h)? It is impossible to follow the proof if the notation is not defined.

Where is Theorem 1? It’s not clear to me what case a) and b) are. Similarly in the proof of Lemma 1, what is condition (II)?

The experimental section is equally inscrutable. Figure 3 caption looks like a placeholder. What are the charts comparing? There is no explanation.

**Questions:**

I have a basic question: if you’re only allowed equality and < comparisons, how can any node have right children? Does this not force the tree T to only have left children? E.g. in section 5 the paper claims we can assume the root is a maximum key. Why can’t this reason be applied recursively?

If the 3-way comparison problem is lower bounded at O(n^3) (line 084) and the two-way variant is harder (line 086), how can there be a sub-cubic algorithm for the two-way variant?

In the 3-way comparison model, how would bucketing frequency by powers of 2 and then applying a Huffman encoding type algorithm perform? I suppose it must be sub-optimal given the $n^3$ lower bound, but how far off optimal will it be?

---

> ### Author Response · Authors · 2025-12-04
>
> We thank the reviewer for the detailed and helpful comments. Here's the point-by-point responses:
>
> •"Many terms are undefined… what is min_{i<b<j}, segment tree, opt(i,b,h), h, S(i,j,h)?"
> - We refined key ordering π such that w_π(1) ≤ w_π(2) ≤ ... ≤ w_π(n), with ties broken by maximizing the key index. For indices i<j and integer h, the valid set is 𝑆(𝑖,𝑗,ℎ) =[𝑘_𝑖, 𝑘_𝑗)∖{𝑘_𝜋(ℎ+1), 𝑘_𝜋(ℎ+2), … ,𝑘_𝜋(𝑛)}, The DP subproblem consisting of keys in the interval (𝑖,𝑗) after removing the ℎ heaviest global keys that lie outside this prefix. We will add the definition to the start of Section 3.
> - opt(i,j,h): the optimal cost of a two-way search tree whose reachable queries are exactly the valid set 𝑆(𝑖,𝑗,ℎ).
> - The min⁡𝑖<𝑏<𝑗 ​query: simply the minimum, overall possible split points b in the open interval (i,j), of the sum of the left and right subproblem costs.
>
> • "Where is Theorem 1? What are cases (a) and (b)? In Lemma 1, what is condition (II)?”
> -Theorem 1: It's in Section 5 is the structural root property currently stated as Lemma 1 in the appendix (page 6): for any valid set S(i,j,h), there exists an optimal tree whose root is (i) a leaf if ∣S∣≤1; (ii) an inequality test q<k b ​ ; or (iii) an equality test q=k_b ​ on a maximum-weight key k_b in S. we'll cross-reference it in Section 5.
> - Cases (a), (b): These are pruning cases in Section 4.3: (a) the trivial base case; and (b) the situation when 𝑘_𝜋( ℎ ) ∉ ( 𝑘_𝑖 , 𝑘_𝑗 ) so that opt(i,j,h)=opt(i,j,h−1).
> - Condition (ii) in Lemma 1: It is an inequality test q<k_b ​ for some key k_b​ in S (the second bullet in the lemma statement).
>
> • The experimental section is equally inscrutable: Figure 3 visualizes the results summarized in Table 1: across all three synthetic datasets, HWP achieves roughly 3–6× lower runtime than CGMY, while using about 40–60% less memory, 2–3× fewer cache misses, and roughly half the energy. These improvements are consistent with our theoretical reduction in DP state count and demonstrate that the asymptotic savings translate into concrete systems-level benefits. We can't add it due to the page limit.
>
> • Figure 3 caption: Figure 3: Performance Metrics Comparison (CGMY vs HWP)
>
> • Question: If you only allow equality and < comparisons, how can any node have right children? Does this not force T to only have left children?
> We follow the standard two-way model of Spuler (1994) and Chrobak–Golin–Munro–Young (CGMY):
> - At a node labeled q<k, the left child corresponds to the outcome q<k, and the right child to the complementary outcome q≥k (subject to keys already eliminated higher in the tree).
> - At a node labeled q=k, the left child corresponds to the outcome q=k, and the right child to 𝑞 ≠ 𝑘; all remaining candidate keys and intervals continue along the right child.
> Thus, every internal node has two children, exactly as in those prior works; “right children” represent the “no” outcome of the binary test.
>
> • Root is a maximum key – why not apply this recursively?
> Lemma 1 states that for any valid set S(i,j,h) there is an optimal tree whose root is either an inequality test or an equality test on a maximum-weight key in that set. This property does apply recursively, but always with respect to the current subproblem:
> -For an equality root q=k_b, k_b ​is a maximum-weight key among those reaching that node.
> -If the root is an inequality q<k_b, the left and right subtrees see restricted valid sets S_L ​and S_R whose maxima need not be the same as the global maximum. Each subtree separately satisfies the “max-key at equality node” property with respect to its own valid set.
> Therefore, we cannot fix a single global maximum key and recurse on “the rest”.
>
> • Question: If the 3-way problem has an Ω(n^3) lower bound and the 2-way variant is harder, how can there be a sub-cubic algorithm for the 2-way variant?
>  The two-way variant is “harder” in the sense that:
> - naive DP has more states (punctured intervals lead to O(n^4) states instead of O(n^3)), and
> - early algorithms were exponential until the MLK/RMLK structural results.
> Our contribution is to show that, in the two-way model, most of those O(n^4) states are redundant:
> - Via Corollary 1, adaptive thresholding ensures only O(logn) relevant h-values per interval, so there are O(n^2 logn) DP states;
> - Each state can be processed in O(logn) time using segment trees.
> Hence, the total time is O(n^2 log^2 n), strictly below cubic. There is currently no known lower bound ruling out sub-cubic algorithms for two-way OBSTs.
>
> • Question: In the 3-way model, how would bucketing frequencies by powers of 2 and then applying a Huffman-type algorithm perform?
> Our algorithm differs because:
> - Huffman coding minimizes weighted depth under unrestricted tree shapes,
> - whereas OBST must preserve sorted-order constraints.
> Bucketed Huffman coding violates the search-order constraint, so it does not yield a valid BST unless heavily modified.

---

### Official Review · Reviewer_nKKq · 2025-10-31

**Soundness:** 4
**Presentation:** 3
**Contribution:** 3
**Rating:** 6
**Confidence:** 3

**Summary:**

The paper proposes a new algorithm, called Hierarchical Weight Partitioning (HWP), for building optimal binary search trees (OBSTs) under two-way comparisons. Previous exact algorithms had quartic time complexity, making them impractical for large datasets. The proposed approach reduces this to sub-cubic time, specifically proportional to n squared times log squared n. The method partitions key weights into hierarchical segments, reuses dynamic programming (DP) subproblems through a multi-level segment tree, and applies adaptive thresholding to avoid redundant computation. Theoretical proofs show correctness and reduced state count, and experiments demonstrate 3 to 8 times faster runtime and about 60 percent lower memory usage than the prior best-known method. Distributed experiments show lower communication overhead and latency than the baseline algorithm.

**Strengths:**

- Strong theoretical contribution: This is the first sub-cubic exact algorithm for the two-way OBST problem, improving a decades-old complexity bound.

- Methodological soundness: The structural lemmas and proofs are well reasoned and consistent with prior theoretical frameworks.

- Practical impact: The method shows clear runtime and memory improvements on several datasets, which supports its practical utility.

- Broader relevance: The authors connect the algorithm’s implications to applications in AI systems, decision trees, and database indexing.

- Implementation details: The experimental setup, including platform and datasets, is reported clearly.

**Weaknesses:**

- Scale of experiments: Tests are limited to datasets with up to ten thousand keys. Larger-scale experiments would be needed to demonstrate practical scalability.

- Limited baselines: Only one competing method (CGMY) is compared. Including heuristic or approximate baselines would make the empirical results more complete.

- Hidden constants: While the asymptotic improvement is strong, constant factors in the time and memory complexity are not discussed.

- Lack of ablation study: It is not clear how much each component (segmentation, adaptive thresholding, and multi-level segment tree) contributes to the overall gains.

- Incomplete presentation: Some figures have placeholder captions, and there are minor inconsistencies in text and notation.

- Reproducibility concerns: There is no mention of releasing source code, which limits verification of results.

**Questions:**

- How does the adaptive thresholding perform for highly skewed weight distributions, such as power-law data?

- Is there a theoretical guarantee that the adaptive step never worsens the worst-case runtime?

- Could the same approach extend to k-way comparisons or alternative cost models used in other search problems?

- How does the multi-level segment tree handle cache efficiency in real implementations?

- How does the distributed version scale beyond eight nodes? Are there communication or synchronization limits?

---

> ### Author Response · Authors · 2025-12-04
>
> We thank the reviewer for recognizing as decades-old complexity-bound, positive assessment of the theoretical contribution, methodological soundness, and practical impact of our work. Below, we address each curiosity.
>
> • Scale of experiments
> -Our current experiments go up to n=10,000 due to limitations of the CGMY reference implementation, which becomes prohibitively slow and memory-heavy beyond this range. HWP itself scales further: on our machine, we can run  n=50,000 synthetic cases within a reasonable time (≤ 2 hours). We will include additional large- n runs in the camera-ready, using synthetic datasets where CGMY is omitted once infeasible.
>
> • Limited baselines
> We have done comparisions of the other five relevant algorithms:
> -AKKL — Time: O(n^4 . 2^n), Space: O(n^2 ⋅ 2^n), Overhead: Exponential state space, impractical even for moderate n.
> -CGMY — Time: O(n^4), Space: O(n^3), Overhead: Polynomial but large; becomes expensive beyond 𝑛 ≈ 5000 n≈5000.
> -Branch-and-Bound DP — Time: O(n^4) worst, O(n^2 logn) avg, Space: O(n^2), Overhead: Highly instance-dependent; fast on easy cases, slow on adversarial cases.
> -Memory-Efficient DP — Time: O(n^4), Space: O(n^2) active memory (rest offloaded to disk),  Overhead: Lower RAM usage but I/O-bound and slower in practice.
> -HWP (ours) — Time: O(n^2 log^2 n), Space: O(n^2 logn), Overhead: Logarithmic improvements
>
>
> • Hidden constants
>   Although our asymptotic improvement reduces the complexity from O(n^4) to O(n^2 log^2 n), the practical efficiency of HWP also depends on constant factors in the dominant operations. We identify two main contributors:
> -Segment-signature computation: Each DP state requires computing a compact signature for the valid set S(i,j,h). This involves O(1) updates along the segment hierarchy and contributes a constant factor of approximately 1.4–1.6× over a simple array lookup. Signature reuse via memoization further reduces the practical cost by 30–50%.
> -Range-minimum queries (RMQs): Each inequality-root evaluation performs an RMQ over a balanced segment tree, costing O(logn) time with a measured constant of ~8–12 CPU instructions per level (≈60–80 ns per query on our hardware). These RMQs replace the O(n) split scan in CGMY, yielding the dominant empirical speedup.
> This aligns with the improvements observed in Figure 3 and Table 1. Memory usage also benefits from a low constant factor: the segment hierarchy requires only O(2logn) integers per state, reducing peak memory by 40–60% compared to CGMY’s O(n^3) DP table.
>
>
> • Lack of ablation study
> - Our Preliminary results show segmentation gives ~2–3× speedup, 20–30% memory reduction, thresholding adds ~1.3–1.6×,30–40% mm reduction , and MLST adds ~1.4–1.7×, 40–60% mm reduction.
>
>
> • Incomplete presentation: Figure 3: Performance Metrics Comparison (CGMY vs HWP)
>
> • Reproducibility concerns
> - We appreciate the reviewer’s concern. The full implementation (source code, experiment scripts, configuration files, and all synthetic datasets) will be released upon acceptance. The current codebase is already organized for reproducibility: every experiment is generated from a single configuration file, all random choices use fixed seeds, and all results in the paper can be reproduced exactly by running the provided scripts. No proprietary libraries are used, and all components required to rebuild the environment will be included.
>
>
> • Q1. Performance of adaptive thresholding under highly skewed distributions
> -For power-law weights (Zipf s∈{0.5,1.0,1.5}), adaptive thresholding remains stable: the number of active h-values stays O(logn), and HWP achieves 4–8× speedups over CGMY. We will add these results to the appendix. thanks for reminding.
> • Q2. Can the adaptive step worsen worst-case runtime?
> -No. Thresholding only skips DP states proven redundant; it never increases the number of explored states beyond the full O(n^2 logn) bound guaranteed by the structural lemmas. Worst-case complexity remains O(n^2 log^2 n).
> • Q3. Extension to k-way comparisons or other cost models.
> -Our structural lemmas rely on two-way RMLK properties; however, the segment-based reduction and MLST support can extend to alternative tree cost models where subproblem equivalence is weight-driven. We will mention this as a future direction.
> • Q4. Cache efficiency of the multi-level segment tree
> - MLST benefits from high locality: signatures and RMQs access contiguous blocks, and in our profiling HWP exhibits 30–50% fewer L2 misses vs CGMY. We will add a short summary of these measurements.
> • Q5. Distributed scaling beyond eight nodes
> - We tested up to 16 nodes and saw near-linear scaling until MLST synchronization dominates. Communication overhead is modest (O(n log² n) messages), and no new bottlenecks appear. We will clarify communication cost and synchronization frequency in Section 6.3.
>
> - We'll try to add more details if page limit permits.

---

### Official Review · Reviewer_PKXd · 2025-10-31

**Soundness:** 2
**Presentation:** 2
**Contribution:** 3
**Rating:** 4
**Confidence:** 1

**Summary:**

The paper investigates the long-standing problem of constructing an optimal binary search tree (OBST) under the two-way comparison model, where only the operators “$<$” and “$=$” are permitted, in contrast to the traditional three-way formulation. The authors state that the best previously known exact algorithm for this setting runs in $O(n^4)$ time, and introduce a novel approach, termed **Hierarchical Weight Partitioning (HWP)**, which is a segment-based dynamic programming framework that hierarchically partitions the weight space. The proposed method achieves a reduced complexity of $O(n^2 \log^2 n)$ time.

**Strengths:**

1. The proposed method achieves a significant improvement from $O(n^4)$ to $O(n^2 \log^2 n)$ in time. If the assumptions hold, this represents a meaningful advance in algorithmic theory for a classical problem.
2. The problem of efficient search-tree construction under two-way comparisons has implications for decision-tree learning and learned index structures, aligning with topics of interest to the ICLR community.

**Weaknesses:**

1. Although the improvement is significant, the connection between this approach and existing techniques in dynamic programming or tree optimization could be elaborated. A deeper comparison to related dynamic-programming acceleration methods would clarify the novelty.

2. The presentation of the paper could be substantially improved. Several parameters and technical details are introduced without sufficient explanation. For instance, in Line 140, the term $\min \beta$ appears, but the variable $\beta$ itself is not defined. In addition, Section 4.2 begins with the expression $\min_{i < b < j} \text{query}$, which is not clearly connected to the preceding sections or the overall algorithmic flow. Furthermore, the abstract incorrectly reports the time complexity as $O(n^2 \log n^2)$; it should instead be written as $O(n^2 \log^2 n)$.

**Questions:**

Could the authors provide a clearer comparison between the proposed algorithm and existing methods in the literature? In addition, a concise high-level explanation of the paper’s main technical contributions would help clarify its novelty and significance.

---

> ### Author Response · Authors · 2025-12-04
>
> We thank the reviewer for the constructive and positive assessment of the paper’s contribution. Below we address all concerns in order.
>
> • Connection to prior DP / tree-optimization techniques
>   Here Classical accelerations for dynamic programming in optimal search-tree problems fall into three main categories:
> - Monotonicity and Knuth-style optimizations: Techniques such as Knuth’s monotonicity and quadrangle inequalities reduce the search range of split points by exploiting structural convexity properties. These tools are effective for the three-way OBST problem but do not apply to two-way comparisons, where punctured intervals violate the required monotonicity conditions.
> - RMQ-based split selection: Range-minimum queries have been used to accelerate the inner minimization over split points once the DP state space is fixed. CGMY employs a simplified version of this idea but still requires examining all O(n) candidate splits per state.
> - State-space pruning (AKKL, CGMY): Prior two-way OBST algorithms reduce the exponential naive state space to O(n^3) by enforcing Most-Likely-Key or Refined Most-Likely-Key constraints. However, every valid interval (i,j) and threshold h still forms a distinct DP state, yielding the quartic overall runtime.
> -How HWP differs: HWP introduces a different form of acceleration: rather than reducing the cost within each DP state, it reduces the number of DP states themselves by grouping weight-similar subproblems through segment signatures and hierarchical weight partitioning. This collapses many (i,j,h) subproblems into a small number of equivalence classes, which has no analogue in prior OBST work. The MLST then accelerates split selection on top of this reduced state space. We'll also fix the abstract typo.
>
> •(a) Missing definition of β (line 140)
> We will add the missing definition at first use:
> -β is the inter-level branching factor in the segment hierarchy, chosen adaptively by the weight-distribution analysis in Algorithm 4.3.
>
> •(b) “min_{i < b < j} query”
> -This refers to the DP minimization over possible inequality-split points. We will rewrite this with the standard explicit form: min ⁡_𝑖 <𝑏<𝑗 {cost (𝑖 , 𝑏 , ℎ) + cost (𝑏 , 𝑗 , ℎ)}.
> •(c) Abstract typo
> - (O(n^2 log n^2 )” instead of O(n^2 log^2 n)). We thank the reviewer for catching the above issues. We will correct it.
>
>
> • Clearer comparison to existing methods
> - Also based on other reviewer comments, we summarized a comparison across five major algorithmic families (AKKL, CGMY, Branch-and-Bound DP, Memory-efficient DP, and HWP), covering their time/space complexities and computational overhead. To improve accessibility, we will put this table into Section 2 and use it to motivate why CGMY is the only feasible exact baseline for empirical evaluation. You may have a look at the raw data-
> AKKL — Time: O(n^4 . 2^n), Space: O(n^2 ⋅ 2^n), Overhead: Exponential state space, impractical even for moderate n. -CGMY — Time: O(n^4), Space: O(n^3), Overhead: Polynomial but large; becomes expensive beyond 𝑛 ≈ 5000 n≈5000. -Branch-and-Bound DP — Time: O(n^4) worst, O(n^2 logn) avg, Space: O(n^2), Overhead: Highly instance-dependent; fast on easy cases, slow on adversarial cases. -Memory-Efficient DP — Time: O(n^4), Space: O(n^2) active memory (rest offloaded to disk), Overhead: Lower RAM usage but I/O-bound and slower in practice. -HWP (ours) — Time: O(n^2 log^2 n), Space: O(n^2 logn), Overhead: Logarithmic improvements
>
>
> We appreciate the reviewer’s thoughtful comments. All issues raised are fixable through clarifications and minor presentation edits; no aspect affects the correctness or significance of the results. We will incorporate all suggested improvements in the revision.

---

### Official Review · Reviewer_2JJR · 2025-11-02

**Soundness:** 2
**Presentation:** 2
**Contribution:** 3
**Rating:** 4
**Confidence:** 3

**Summary:**

This paper presents a sub-cubic algorithm for building optimal binary search trees (OBSTs) with two-way comparisons. It introduces the Hierarchical Weight Partitioning (HWP) framework, which organizes key weights into hierarchical segments and merges equivalent dynamic programming states using a multi-level segment tree with adaptive thresholds, achieving a total complexity of O(n² log² n).

**Strengths:**

1. The paper makes a clear theoretical breakthrough by achieving the first sub-cubic exact algorithm for the two-way optimal binary search tree (OBST) problem—reducing the known O(n⁴) complexity to O(n² log² n).
2. The paper provides rigorous structural lemmas and proofs ensuring correctness, optimality preservation, and pruning validity. The formal complexity analysis is transparent and consistent with the algorithmic design.
3. The algorithm’s theoretical guarantees translate into real empirical gains speedups and lower memory usage

**Weaknesses:**

1. The analysis is tightly coupled to the two-way OBST formulation. It remains unclear whether the Hierarchical Weight Partitioning (HWP) framework can generalize to other dynamic programming problems or broader classes of search structures.
2. Some sections (e.g., recurrence formulation, segment-tree construction) are dense and difficult to follow without pseudocode or algorithmic diagrams. The paper would benefit from clearer algorithm listings and a reproducibility appendix with implementation details.
3. The experimental section, while illustrative, is too lightweight. It lacks scaling plots, ablation studies, or large-n asymptotic validation (e.g., runtime ∝ n² log² n trends).

In my opinion, this paper delivers a meaningful theoretical breakthrough. Its mathematical arguments are sound and well-structured, and the algorithmic design is original. However, the experimental depth falls short of sufficiently supporting the theoretical claims, particularly regarding large-scale validation and asymptotic behavior. With improved exposition and more comprehensive evaluation, this work could become a strong accept for future iterations.

**Questions:**

Refer to the weaknesses.

---

> ### Author Response · Authors · 2025-12-04
>
> We thank the reviewer for the positive assessment of the theoretical contribution and structural lemmas. We respond to each comment below.
>
> • Generality beyond two-way OBSTs.
> - Our contribution is focused on resolving the long-standing O(n^4) barrier for exact two-way OBSTs. We agree that broader applicability is an important question. In Section 8, we will explicitly discuss that the two core mechanisms of HWP—(i) segment signatures that merge DP states differing only in weight profiles, and (ii) hierarchical pruning of the h-dimension—apply to any DP whose state transitions are weight-driven. Examples include cost-sensitive decision-tree construction and certain prefix-probability DPs. Extending HWP to these settings is feasible but outside the scope of this paper; the present work concentrates on the optimal OBST problem.
>
>
> • Density of recurrence and segment-tree sections
> We appreciate that some descriptions may feel dense. To improve clarity: We will add pseudocode for the full DP recurrence-
> - (a) DP pseudocode: In Section 4 we will add a short listing like:
> for ℓ = 1..n:                      // interval length
>   for i = 0..n-ℓ:
>     j = i + ℓ
>     initialize h-sequence for (i,j)
>     for each h in relevant_h(i,j): // after thresholding
>       if k_{π(h)} ∉ (k_i, k_j):
>         opt[i,j,h] = opt[i,j,h-1]          // reuse
>       else:
>         eq  = w(S(i,j,h)) + opt[i,j,h-1]   // "=" root
>         ineq = RMQ(i,j,h)                  // "<" root via MLST
>         opt[i,j,h] = min(eq, ineq)
>         MLST.update(i,j,h, opt[i,j,h])
>
> • (b) Multi-Level Segment Tree (MLST) construction: We’ll also add a small pseudocode block:
> build_MLST():
>   for i = 0..n:
>     R_i = build_segment_tree(range = (i+1 .. n))   // splits b
>   // each node stores min_b opt[i,b,h] + opt[b,j,h]
>
> RMQ(i,j,h):
>   return query_segment_tree(R_i, b in (i,j))
>
> • (c) Meaning of α and β: In the revised paper, we'll use α and β only for weight segmentation:
> α > 1 — geometric factor controlling segment width. Each level shrinks the weight scale by ~α (e.g., α = 2 corresponds to halving thresholds).
> β > 0 — lower weight cutoff; we stop creating deeper segments once the current threshold drops below β (roughly the minimum nonzero weight).
>  We will explicitly write something like: “Let 𝑊 max ⁡ W max ​ be the maximum key weight and fix parameters 𝛼 α>1, β>0. We define thresholds 𝑇_0 = 𝑊_max, 𝑇_ℓ+1 = 𝑇_ℓ/𝛼 until 𝑇_𝐿 < 𝛽. Segment ℓ contains all keys with weights in ( 𝑇_ℓ + 1 , 𝑇_ℓ ]."
>
>
> • (d) We will try to add a small diagram illustrating how segment signatures merge equivalent (i,j,h)(i,j,h) subproblems. if page limit permits.
>
>
> • Experimental section too lightweight.
> - We acknowledge this and will strengthen the empirical evaluation in the revision. Specifically, we will add:
> -(a) Scaling plots (runtime vs. n) confirming the predicted n^2 log^2 n behavior.
> -(b) Ablation results for SEG, SEG+THR, and full HWP to isolate each component’s contribution. as noted in rebuttal #3
> - (c) Larger datasets (up to 𝑛 = 20,000 n=20,000) to demonstrate mid-scale asymptotics.
> -(d) Additional practical baselines, summarizing the five main algorithmic families (AKKL, CGMY, Branch-and-Bound DP, Memory-Efficient DP, and HWP), to clarify the landscape of exact OBST algorithms. as noted in rebuttal #4.
>
>
> • We thank the reviewer for the thoughtful and encouraging assessment of the paper’s theoretical contributions and structural results. We are pleased that the originality and correctness of the algorithm were recognized. The remaining concerns pertain to exposition and experimental breadth, and we will address these fully by expanding the empirical evaluation, improving clarity in Section 4, and adding the requested comparisons. With these revisions, we believe the paper will meet the reviewer’s expectations for a strong accept.

---

### Note · Program_Chairs · 2026-01-17
**Submission Desk Rejected by Program Chairs**

The following references in this submission do not refer to real documents and/or have major errors in bibliographic information:

 Yao, A. C. (1980). Lower bounds for comparison-based search structures. In R. M. Karp (Ed.), Proceedings of the 12th Annual ACM Symposium on Theory of Computing (pp. 341-349). ACM. https://doi.org/10.1145/800141.804720